# Association of N, N-diethyl-m-toluamide (DEET) with arthritis among adult participants

Taihong Lv[1☯], Hanming Yu[2☯], Zishuo Ji[3], Yuru Chen[1], Qi Zhao[1], Li Ma [1]*

1 Department of General Medicine, Beijing TianTan Hospital, Capital Medical University, Beijing, China,
2 Department of Pulmonary and Critical Care Medicine, The Fourth Affiliated Hospital of China Medical University, Shenyang, China, 3 Department of Neurology, Beijing TianTan Hospital, Capital Medical University, Beijing, China

☯ These authors contributed equally to this work.
* mali_ttyy@126.com

## Abstract

### Background

N, N-diethyl-m-toluamide (DEET), an active ingredient prevalent in insect repellents, has its effects on human health under ongoing debate and scrutiny.

### Objective

This study aimed to investigate the association between exposure to DEET and arthritis outcomes within the broader adult demographic, leveraging data obtained from the National Health and Nutrition Examination Survey (NHANES).

### Methods

3-diethyl-carbamoyl benzoic acid (DCBA) was used as a specific indicator of DEET exposure. Drawing on NHANES 2007–2016 data, our study incorporated 7065 adults to examine urinary DCBA impacts on arthritis risks using logistic regression and cubic spline analysis.

### Results

Our study included a total of 7065 patients, of which 1860 (26.33%) had arthritis. After adjusting for all covariates, the multivariate logistic regression analysis showed that high DCBA levels (>7.37 ug/L) were associated with an increased likelihood of arthritis (OR: 1.236, 95% CI: 1.037–1.474, $p = 0.018$). Nevertheless, participants in the highest quartile of DCBA levels (Q4) were associated with a 33.9% decreased risk of rheumatoid arthritis (OR: 0.661; 95% CI: 0.501–0.872; $p = 0.003$).

### Conclusion

It was observed that increased levels of DCBA are positively associated with the prevalence of arthritis in the adult population. Conversely, high concentrations of DCBA showed a reverse correlation with the prevalence of rheumatoid arthritis.

**Data Availability Statement:** The dataset underpinning this investigation was sourced from the National Health and Nutrition Examination Survey (NHANES), publicly available at the official

NHANES web portal: https://www.cdc.gov/nchs/nhanes/index.htm.

**Funding:** The author(s) received no specific funding for this work.

**Competing interests:** The authors have declared that no competing interests exist.

**Abbreviations:** BMI, Body mass index; CI, Confidence interval; DCBA, 3-diethyl-carbamoyl benzoic acid; DEET, N, N-Diethyl-m-toluamide; NHANES, National Health and Nutrition Examination Survey; RA, Rheumatoid arthritis.

# 1 Introduction

Arthritis encompasses a spectrum of joint disorders characterized by inflammation, which is frequently associated with pain and progressive joint degeneration [1]. Rheumatoid arthritis (RA) constitutes the predominant variant of arthritic conditions [2]. Rheumatoid arthritis represents a systemic autoimmune disorder marked by synovial membrane inflammation, exerting significant individual and societal impacts [3]. The global age-standardized prevalence and prevalence rates of rheumatoid arthritis rose by 7.4% and 8.2% respectively since 1990, reaching 246.6 and 14.9 per 100,000 in 2017 [4]. The elevated prevalence of rheumatoid arthritis in developed nations is often attributed to a confluence of environmental factors, genetic predisposition, and more comprehensive reporting practices [5–7]. Nevertheless, untreated rheumatoid arthritis can culminate in substantial morbidity, impaired functional capacity, and abbreviated life expectancy [8, 9].

N,N-Diethyl-m-toluamide (DEET) serves as the principal component in numerous insect repellent formulations and has become the predominant and most efficacious repellent across the United States [10–12]. Since its establishment in 1946, N,N-diethyl-meta-toluamide, commonly referred to as DEET, has risen to prominence as a widely adopted insect repellent in the United States [13, 14]. Roughly 30% of the U.S. populace annually employs at least one product infused with DEET [15]. The substance plays a pivotal role in public health due to its effectiveness in warding off vectors such as insects and ticks, which are known to transmit a vast array of infectious diseases [16, 17]. In contemporary times, an assortment of DEET-infused products is accessible to the public, spanning various modalities like liquids, lotions, sprays, and even saturated materials, for instance, towelettes and roll-on dispensers [18]. DEET plays an essential role in public health by aiding in the prevention of arthropod bites from vectors like mosquitoes, ticks, and fleas [19]. With the market offering upwards of 225 insect repellent brands featuring DEET concentrations varying from 4% to 100%, human exposure to DEET is anticipated [18]. Recent studies have highlighted the potential neurotoxicity and cardiotoxicity of DEET, which may result in kidney stones, hyperuricemia, cardiovascular diseases and obesity [20–23]. A major oxidative metabolite of DEET, DCBA, is a sensitive and specific indicator of DEET exposure [15].

Rheumatoid arthritis represents a significant global public health issue, with age-standardized prevalence and prevalence rates on an upward trajectory [4, 6]. To date, there has been a notable research gap regarding the use of the NHANES database to interrogate the relationship between DEET and arthritis; hence, this cross-sectional study delves into the potential connection between DCBA levels and the prevalence of arthritis within the U.S. population, utilizing data spanning the 2007–2016 NHANES. We posited that individuals with arthritis would exhibit elevated concentrations of DCBA.

# 2 Materials and methods

## 2.1. Data source and participants

In compliance with the approved protocol by the National Center for Health Statistics Research Ethics Review Board, participants have provided their written informed consent for participation in NHANES. The dataset employed in this study is anonymized, in the public domain, and was utilized without the need for further ethical clearance from the Swedish Ethical Review Authority, adhering to the national guidelines for research.

Our study drew upon data garnered from the NHANES-an organized, multi-tiered probability initiative capturing a representative cross-section of the non-institutionalized civilian

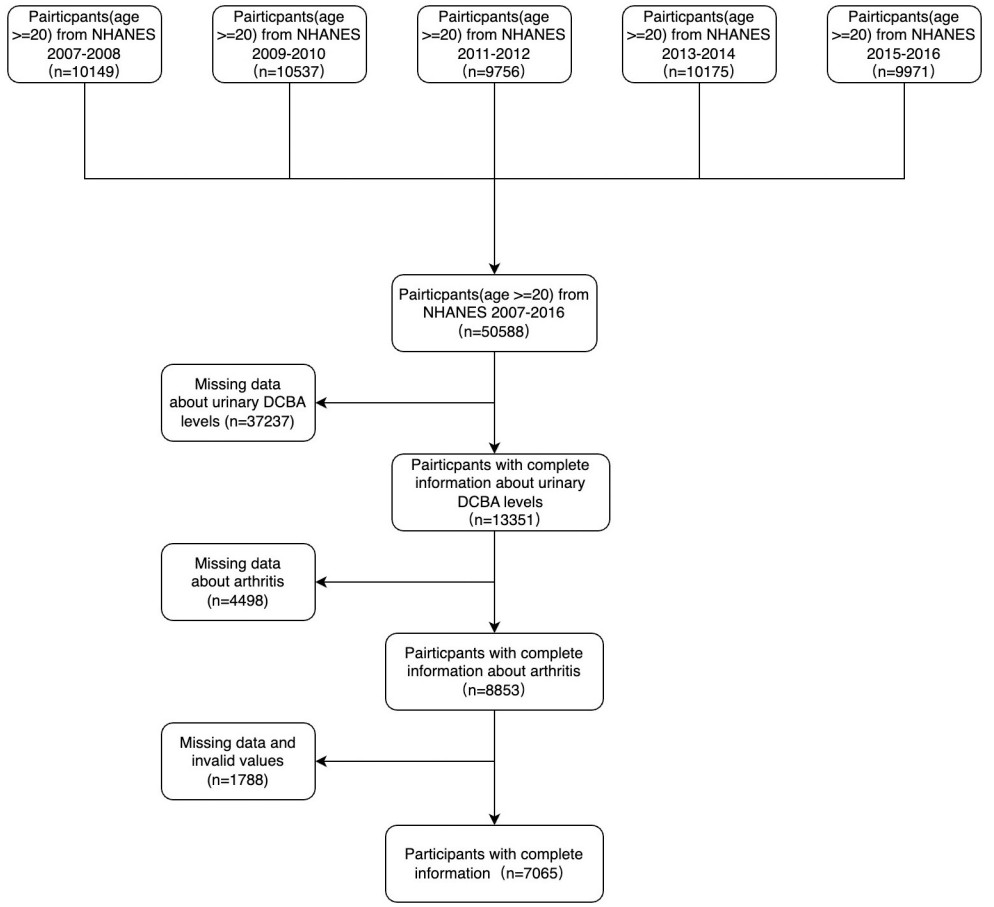

**Fig 1. Flowchart of the population included in our final analysis.**

demographic within the United States [24]. Pertinent details concerning NHANES are available at https://www.cdc.gov/nchs/nhanes/index.htm.

The methodological protocols utilized for the NHANES were sanctioned by the Institutional Review Board (IRB) of the National Center for Health Statistics, functioning under the aegis of the Centers for Disease Control and Prevention. In compliance with ethical standards, informed consent was appropriately obtained from all subjects prior to their inclusion in the study. Comprehensive outlines of the methodologies and consent documentation are available for review at https://cdc.gov/nchs/nhanes/irba98.htm.

Fig 1 delineates the research design, sampling, and exclusion procedures employed in the study. The research encompassed two publicly accessible datasets from the survey cycles of 2007–2016. In each cycle, subjects were selected based on the following criteria: 1) Adolescents under the age of 20 were excluded; 2) Participants lacking DEET analysis were omitted; 3) Individuals without a diagnosis of arthritis were excluded; 4) Data missing multiple covariates, such as age, gender, race, education level, marital status, income, smoking, alcohol consumption, physical activity, sleep disorders, hypertension, and diabetes were also excluded from the study. Across both cycles, a combined total of 7065 participants were included in our analysis.

## 2.2. Measures

**2.2.1. DCBA levels.** Urine specimens were cryopreserved at a temperature of -20°C and subsequently transported to the Division of Environmental Health Laboratory Sciences within the CDC's National Center for Environmental Health. Analytical quantification of DEET and its two derivatives, 3-(diethyl carbamoyl) benzoic acid (DCBA) and N, N-diethyl-3-(hydroxy-methyl)benzamide (DHMB), was conducted via solid-phase extraction followed by high-performance liquid chromatography-tandem mass spectrometry. Considering the relatively low detectable levels of DEET and DHMB, only the concentration of DCBA was incorporated into our analysis. The detection threshold for DCBA was established at 0.929 μg/L for NHANES 2007–2008 and at 0.475 μg/L for subsequent cycles, namely NHANES 2009–2010, 2011–2012, 2013–2014, and 2015–2016.

**2.2.2. Arthritis.** Arthritis diagnosis was ascertained through a self-administered questionnaire (MCQ160a), wherein participants were queried, "Has a doctor or other health professional ever informed you that you have arthritis?" with the possibilities for response being "Yes" or "No." Assessment of rheumatoid arthritis involved a supplementary inquiry: "Which type of arthritis was diagnosed?" Participants could select from "Rheumatoid arthritis," "Osteoarthritis," "Psoriatic arthritis," "Other," or indicate "Refused" or "Don't know." A previous investigation has documented a high degree of concordance (85%) between self-reported arthritis and arthritis verified by clinical diagnosis.

## 2.3. Covariates

Building upon existing scholarship, a variety of potential covariates linked with arthritis have been cataloged in the literature [25–27]. To discern the extent of mediating effects, present among these covariates, we employed the analytical framework introduced by Baron and Kenny [28]. The spectrum of covariates examined includes socio-demographic factors, behavioral attributes, and health-related characteristics. Within the socio-demographic domain, variables consisted of age brackets (20–44 years, 45–59 years, and 60–80 years), gender (female and male), race (Non-Hispanic White, Mexican American, Non-Hispanic Black, Other Hispanic and Other Race), marriage (encompassing categories of Married, Widowed, Divorced, Separated, Never married, and Living with partner), educational levels (categorized as less than 9th grade, 9-11th grade, high school graduate, some college or AA degree and college graduate or above), and economic standing as assessed by the poverty income ratio (PIR), with the threshold for poverty defined as a PIR under 1 for the respective household.

Behavioral attributes were defined by smoking habits (participants who reported consuming 100 or more cigarettes in their lifetime were identified as smokers, which was recorded dichotomously as yes or 'no), alcohol use (no or yes), and the degree of physical activity (inactive or active).

Health-related determinants included body mass index (BMI < 24 indicating underweight/normal weight and BMI > = 24 indicating overweight/obesity), sleep disorders (no or yes), the occurrence of hypertension (no or yes), and diabetes status (no or yes).

## 2.4. Statistical analysis

In our investigation, a cross-sectional methodology was employed to examine data sourced from adult subjects in the National Health and Nutrition Examination Survey (NHANES), spanning the years 2007 to 2016. To quantitatively assess exposure to DEET, we utilized DCBA as a bio-indicator. The distribution of DCBA among the study cohort was segregated into quartiles. Using multiple logistic regression analyses, we investigated the association between levels of DCBA and the prevalence of arthritis in the adult population. Stratified

**Table 1. Characteristics of participants across NHANES 2007–2016 cycles.**

| Variables | Classification | N | Year | | | | | Total | p |
|---|---|---|---|---|---|---|---|---|---|
| | | | 2007–2008 | 2009–2010 | 2011–2012 | 2013–2014 | 2015–2016 | | |
| DCBA | Q1 (<0.682) | n | 404 | 234 | 371 | 460 | 300 | 1769 | <0.001 |
| | | % | 22.80% | 13.20% | 21.00% | 26.00% | 17.00% | 100.00% | |
| | Q2 (0.682–2.03) | n | 303 | 420 | 333 | 412 | 301 | 1769 | |
| | | % | 17.10% | 23.70% | 18.80% | 23.30% | 17.00% | 100.00% | |
| | Q3 (2.03–7.37) | n | 384 | 380 | 322 | 338 | 337 | 1761 | |
| | | % | 21.80% | 21.60% | 18.30% | 19.20% | 19.10% | 100.00% | |
| | Q4 (>7.37) | n | 368 | 474 | 262 | 282 | 380 | 1766 | |
| | | % | 20.80% | 26.80% | 14.80% | 16.00% | 21.50% | 100.00% | |
| Arthritis | no | n | 1053 | 1096 | 972 | 1084 | 1000 | 5205 | 0.07 |
| | | % | 20.20% | 21.10% | 18.70% | 20.80% | 19.20% | 100.00% | |
| | yes | n | 406 | 412 | 316 | 408 | 318 | 1860 | |
| | | % | 21.80% | 22.20% | 17.00% | 21.90% | 17.10% | 100.00% | |

analysis was employed to evaluate the consistency of these associations across heterogeneous populations. Stratification was applied to the dataset considering a range of critical demographics, encompassing age, sex, and race.

We subsequently delved deeper into the association between DCBA concentrations and the prevalence of rheumatoid arthritis in adults utilizing multivariate logistic regression to dissect this link.

Furthermore, we employed Restricted Cubic Splines (RCS) regression to explore the dose-response curve relating DCBA exposure to arthritis prevalence.

## 3 Results

Table 1 delineates the baseline demographics of the study cohort throughout the different survey waves. The calculated prevalence rate of arthritis stood at 26.33%. However, analyses across the five cycles revealed no statistically significant fluctuations ($p = 0.07$). In stark contrast, significant shifts in the distribution of DCBA levels were noted over the successive cycles ($p < 0.001$).

Participant characteristics, stratified by the presence or absence of arthritis, are detailed in Table 2. Univariate analyses suggest that family income and BMI exhibit no association with arthritis prevalence. Conversely, factors such as age, gender, ethnicity, educational attainment, marital status, smoking status, alcohol consumption, physical activity, sleep disorders, hypertension, diabetes, and DCBA levels may potentially serve as risk determinants for arthritis among the adult population in the United States.

Table 3 presents the outcomes of binary logistic regression analyses assessing the impact of DCBA levels on arthritis risk. When adjustments were made for a comprehensive set of socio-demographic, behavioral, and health-related covariates, DCBA levels within the second and third quartiles (Q2, Q3) showed no significant association with arthritis risk among American adults. Conversely, individuals in the highest quartile of DCBA levels (Q4) demonstrated a 23.6% elevated risk of arthritis (OR: 1.236; 95% CI: 1.037–1.474; $p = 0.018$) compared to those in the lowest quartile (Q1).

In Fig 2, subgroup analyses were performed to further explore whether subgroups had differential effects on the association of exposure to DCBA with arthritis. Significantly, a robust positive correlation between serum DCBA levels and arthritis risk was evident across most

**Table 2. Characteristics of participants with/without the arthritis.**

| Variables | Classification | N | Arthritis | | Total | p |
|---|---|---|---|---|---|---|
| | | | 0 | 1 | | |
| DCBA | Q1 (<0.682) | n | 1280 | 489 | 1769 | 0.045 |
| | | % | 72.40% | 27.60% | 100.00% | |
| | Q2 (0.682–2.03) | n | 1300 | 469 | 1769 | |
| | | % | 73.50% | 26.50% | 100.00% | |
| | Q3 (2.03–7.37) | n | 1341 | 420 | 1761 | |
| | | % | 76.10% | 23.90% | 100.00% | |
| | Q4 (>7.37) | n | 1284 | 482 | 1766 | |
| | | % | 72.70% | 27.30% | 100.00% | |
| Age | 20–44 | n | 2821 | 235 | 3056 | <0.001 |
| | | % | 92.30% | 7.70% | 100.00% | |
| | 45–59 | n | 1253 | 497 | 1750 | |
| | | % | 71.60% | 28.40% | 100.00% | |
| | 60–80 | n | 1131 | 1128 | 2259 | |
| | | % | 50.10% | 49.90% | 100.00% | |
| Gender | male | n | 2737 | 731 | 3468 | <0.001 |
| | | % | 78.90% | 21.10% | 100.00% | |
| | female | n | 2468 | 1129 | 3597 | |
| | | % | 68.60% | 31.40% | 100.00% | |
| Race | Mexican American | n | 846 | 187 | 1033 | <0.001 |
| | | % | 81.90% | 18.10% | 100.00% | |
| | Other Hispanic | n | 544 | 170 | 714 | |
| | | % | 76.20% | 23.80% | 100.00% | |
| | Non-Hispanic White | n | 2132 | 1008 | 3140 | |
| | | % | 67.90% | 32.10% | 100.00% | |
| | Non-Hispanic Black | n | 1073 | 383 | 1456 | |
| | | % | 73.70% | 26.30% | 100.00% | |
| | Other Race | n | 610 | 112 | 722 | |
| | | % | 84.50% | 15.50% | 100.00% | |
| Education | Less than 9th grade | n | 445 | 207 | 652 | 0.005 |
| | | % | 68.30% | 31.70% | 100.00% | |
| | 9-11th grade | n | 715 | 273 | 988 | |
| | | % | 72.40% | 27.60% | 100.00% | |
| | High school graduate | n | 1126 | 435 | 1561 | |
| | | % | 72.10% | 27.90% | 100.00% | |
| | Some college or AA degree | n | 1575 | 592 | 2167 | |
| | | % | 72.70% | 27.30% | 100.00% | |
| | College graduate or above | n | 1344 | 353 | 1697 | |
| | | % | 79.20% | 20.80% | 100.00% | |

(*Continued*)

**Table 2.** (Continued)

| Variables | Classification | N | Arthritis | | Total | p |
|---|---|---|---|---|---|---|
| | | | 0 | 1 | | |
| Marriage | Married | n | 2689 | 962 | 3651 | <0.001 |
| | | % | 73.70% | 26.30% | 100.00% | |
| | Widowed | n | 220 | 282 | 502 | |
| | | % | 43.80% | 56.20% | 100.00% | |
| | Divorced | n | 509 | 289 | 798 | |
| | | % | 63.80% | 36.20% | 100.00% | |
| | Separated | n | 169 | 70 | 239 | |
| | | % | 70.70% | 29.30% | 100.00% | |
| | Never married | n | 1159 | 185 | 1344 | |
| | | % | 86.20% | 13.80% | 100.00% | |
| | Living with partner | n | 459 | 72 | 531 | |
| | | % | 86.40% | 13.60% | 100.00% | |
| Diabetes | yes | n | 462 | 418 | 880 | <0.001 |
| | | % | 52.50% | 47.50% | 100.00% | |
| | no | n | 4743 | 1442 | 6185 | |
| | | % | 76.70% | 23.30% | 100.00% | |
| Hypertension | yes | n | 1418 | 1127 | 2545 | <0.001 |
| | | % | 55.70% | 44.30% | 100.00% | |
| | no | n | 3787 | 733 | 4520 | |
| | | % | 83.80% | 16.20% | 100.00% | |
| Drink | yes | n | 3820 | 1270 | 5090 | <0.001 |
| | | % | 75.00% | 25.00% | 100.00% | |
| | no | n | 1385 | 590 | 1975 | |
| | | % | 70.10% | 29.90% | 100.00% | |
| Smoke | yes | n | 2176 | 996 | 3172 | <0.001 |
| | | % | 68.60% | 31.40% | 100.00% | |
| | no | n | 3029 | 863 | 3892 | |
| | | % | 77.80% | 22.20% | 100.00% | |
| Physical activity | yes | n | 1984 | 616 | 2600 | 0.007 |
| | | % | 76.30% | 23.70% | 100.00% | |
| | no | n | 3221 | 1244 | 4465 | |
| | | % | 72.10% | 27.90% | 100.00% | |
| Sleep disorders | yes | n | 1007 | 817 | 1824 | <0.001 |
| | | % | 55.20% | 44.80% | 100.00% | |
| | no | n | 4198 | 1043 | 5241 | |
| | | % | 80.10% | 19.90% | 100.00% | |
| BMI | <24 | n | 1343 | 294 | 1637 | 0.133 |
| | | % | 82.00% | 18.00% | 100.00% | |
| | >=24 | n | 3862 | 1566 | 5428 | |
| | | % | 71.10% | 28.90% | 100.00% | |
| Income | yes | n | 4248 | 1547 | 5795 | 0.152 |
| | | % | 73.30% | 26.70% | 100.00% | |
| | no | n | 957 | 313 | 1270 | |
| | | % | 75.40% | 24.60% | 100.00% | |

**Table 3. Associations of DCBA levels with arthritis in American adults.**

|  | classification | model1 | | | | model2 | | | | model3 | | | |
|---|---|---|---|---|---|---|---|---|---|---|---|---|---|
|  |  | p | or | 95%CI | | p | or | 95%CI | | p | or | 95%CI | |
|  |  |  |  | LL | UL |  |  | LL | UL |  |  | LL | UL |
| DCBA | Q1(<0.682) | - | ref | ref | ref | - | ref | ref | ref | - | ref | ref | ref |
|  | Q2 (0.682–2.03) | 0.406 | 1.074 | 0.908 | 1.271 | 0.383 | 1.078 | 0.91 | 1.277 | 0.604 | 1.048 | 0.879 | 1.249 |
|  | Q3 (2.03–7.37) | 0.833 | 1.019 | 0.858 | 1.21 | 0.815 | 1.021 | 0.859 | 1.213 | 0.777 | 0.975 | 0.815 | 1.165 |
|  | Q4 (>7.37) | 0.002 | 1.308 | 1.104 | 1.549 | 0.003 | 1.291 | 1.089 | 1.53 | 0.018 | 1.236 | 1.037 | 1.474 |

subgroups. Importantly, the described association did not reach statistical significance within the subgroup comprising non-Mexican Hispanic males between the ages of 45–59.

Table 4 presents the outcomes of binary logistic regression analyses assessing the impact of DCBA levels on rheumatoid arthritis risk. When adjustments were made for a comprehensive set of sociodemographic, behavioral, and health-related covariates, DCBA levels within the second quartiles (Q2) showed no significant association with rheumatoid arthritis risk among American adults. In contrast, individuals within the third quartiles of DCBA levels (Q3) exhibited a 31.9% reduction in rheumatoid arthritis risk (OR: 0.681; 95% CI: 0.514–0.903; *p* = 0.008)

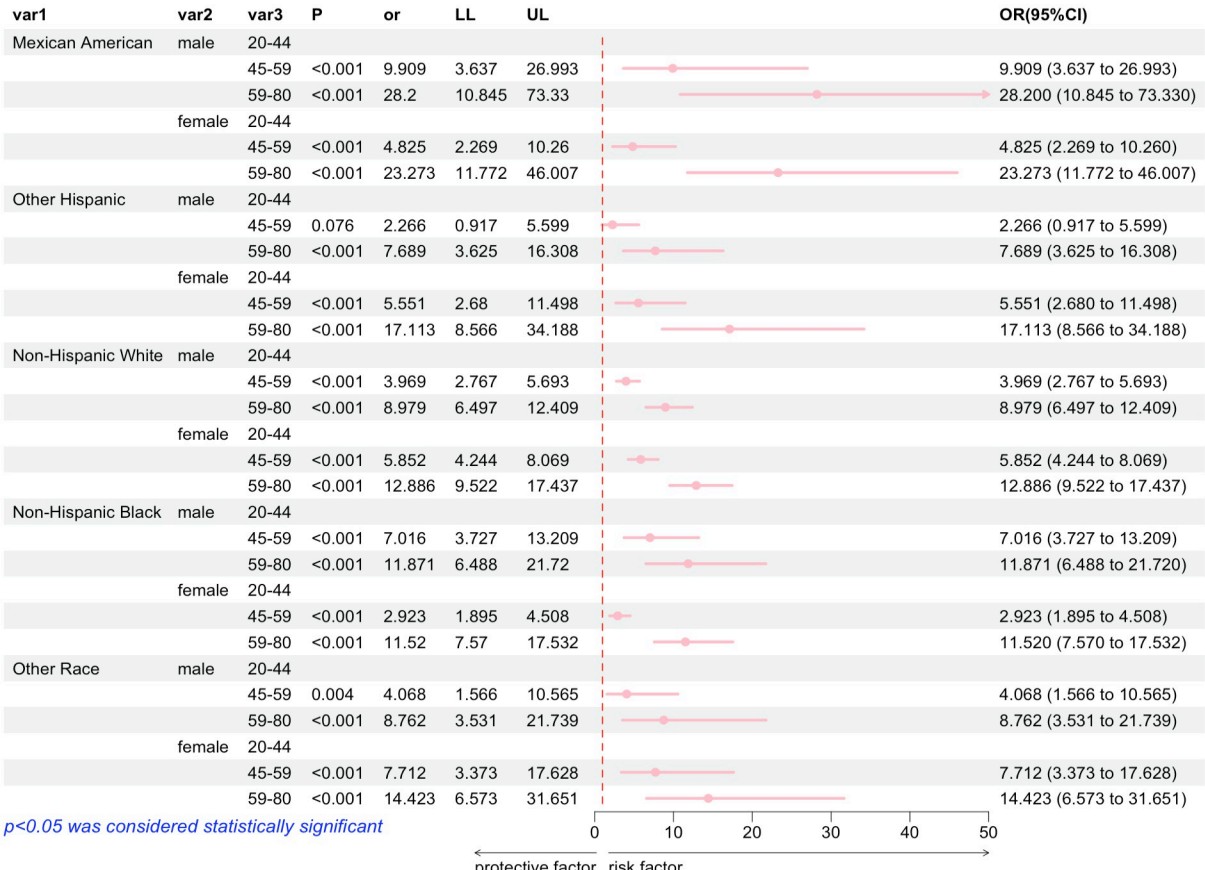

**Fig 2. Stratified analyses of the associations between quartile s of urinary DCBA levels with the prevalence of arthritis among the general adult population.**

**Table 4. Associations of DCBA levels with rheumatoid arthritis in American adults.**

| | classification | model1 | | | | model2 | | | | model3 | | | |
|---|---|---|---|---|---|---|---|---|---|---|---|---|---|
| | | p | or | 95%CI | | p | or | 95%CI | | p | or | 95%CI | |
| | | | | LL | UL | | | LL | UL | | | LL | UL |
| DCBA | Q1(<0.682) | - | ref | ref | ref | - | ref | ref | ref | - | ref | ref | ref |
| | Q2 (0.682–2.03) | 0.102 | 0.799 | 0.61 | 1.046 | 0.102 | 0.798 | 0.609 | 1.046 | 0.093 | 0.793 | 0.605 | 1.039 |
| | Q3 (2.03–7.37) | 0.008 | 0.681 | 0.514 | 0.903 | 0.007 | 0.68 | 0.512 | 0.901 | 0.008 | 0.682 | 0.514 | 0.906 |
| | Q4 (>7.37) | 0.003 | 0.661 | 0.503 | 0.87 | 0.003 | 0.659 | 0.5 | 0.868 | 0.003 | 0.661 | 0.501 | 0.872 |

**Table 5. Associations of DCBA levels with non-rheumatoid arthritis in American adults.**

| | classification | model1 | | | | model2 | | | | model3 | | | |
|---|---|---|---|---|---|---|---|---|---|---|---|---|---|
| | | p | or | 95%CI | | p | or | 95%CI | | p | or | 95%CI | |
| | | | | LL | UL | | | LL | UL | | | LL | UL |
| DCBA | Q1(<0.682) | - | ref | ref | ref | - | ref | ref | ref | - | ref | ref | ref |
| | Q2 (0.682–2.03) | 0.102 | 1.252 | 0.956 | 1.64 | 0.102 | 1.253 | 0.956 | 1.641 | 0.093 | 1.261 | 0.962 | 1.653 |
| | Q3 (2.03–7.37) | 0.008 | 1.468 | 1.108 | 1.946 | 0.007 | 1.472 | 1.11 | 1.951 | 0.008 | 1.465 | 1.104 | 1.945 |
| | Q4 (>7.37) | 0.003 | 1.512 | 1.149 | 1.99 | 0.003 | 1.518 | 1.152 | 1.999 | 0.003 | 1.512 | 1.147 | 1.994 |

relative to those in the lowest quartile (Q1). Similarly, participants in the highest quartile of DCBA levels (Q4) were associated with a 33.9% decreased risk of rheumatoid arthritis (OR: 0.661; 95% CI: 0.503–0.87; $p = 0.003$) in comparison to their counterparts in the lowest quartile (Q1).

Table 5 details the results from binary logistic regression analyses regarding the influence of varying DCBA concentrations on the susceptibility to non-rheumatoid arthritis. Controlling for the same covariates, the transition from the second DCBA quantiles (Q2) demonstrated an absence of a significant relationship with the risk of non-rheumatoid arthritis among the American adult populace. In stark contrast, subjects within the intermediary DCBA quantiles (Q3) encountered a pronounced 46.5% escalation in the likelihood of non-rheumatoid arthritis (OR: 1.465; 95% CI: 1.104–1.945; $p = 0.008$), when juxtaposed with individuals in the lowest quartile (Q1). In a similar vein, the participants categorized in the uppermost quartile of DCBA exposure (Q4) bore a 51.2% augmented risk of developing non-rheumatoid arthritis (OR: 1.512; 95% CI: 1.147–1.994; $p = 0.003$), in comparison to those situated in the lowest quartile.

Upon the exclusion of anomalies, subsequent analyses incorporating RCS were conducted to expound upon the dose-response relationship between DCBA concentrations and the risk of developing arthritis. Fig 3A illustrates a positive linear relationship between DCBA concentration and the prevalence of arthritis, albeit without evidence of non-linearity ($p = 0.413$). Conversely, Fig 3B depicts a negative association between DCBA levels and the occurrence of rheumatoid arthritis, with statistically significant non-linear trends ($p = 0.002$). Similarly, Fig 3C demonstrates a positive linear association between DCBA levels and the prevalence of rheumatoid arthritis, also presenting significant non-linear patterns ($p = 0.002$).

## 4 Discussion

To our knowledge, this is the first report to describe an association between DEET exposure and arthritis based on a large-scale dataset with high-quality data on human exposure to DEET. In this cross-sectional study of pooled NHANES data from 2007 to 2016 and involving

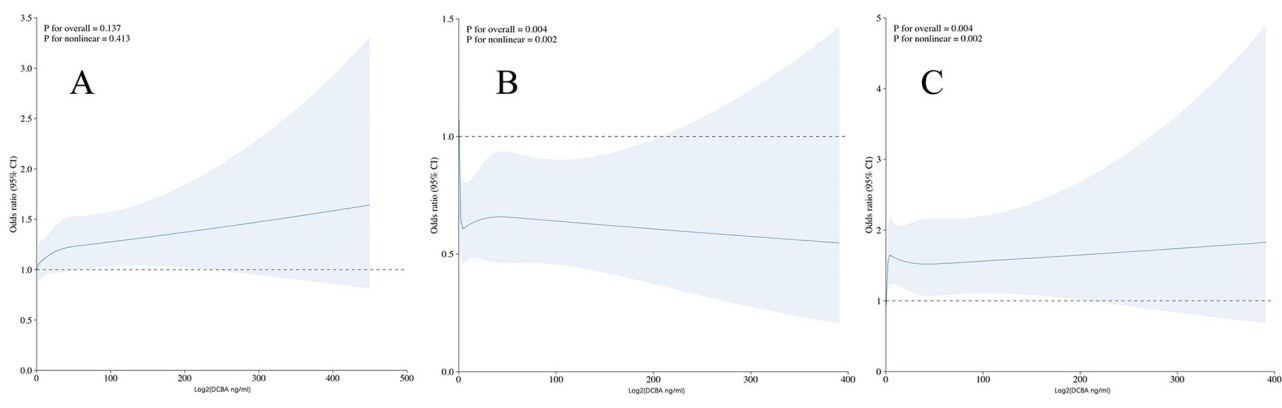

**Fig 3. Restricted cubic spline model.**

a general population of 7650 adults in the United States, the urinary concentration of DCBA, a major metabolite of DEET, was significantly and positively associated with non-rheumatoid arthritis.

Our research findings indicate that significantly higher concentrations of DCBA in adults diagnosed with arthritis compared to those without arthritis ($p = 0.045$). In the quartile with the highest urinary DCBA levels, there was an increased prevalence of arthritis, especially non-rheumatoid arthritis (OR: 1.512; $p = 0.003$), whereas there was a negative correlation with the risk of developing rheumatoid arthritis (OR: 0.661; $p = 0.003$). Furthermore, in other quartiles, urinary DCBA levels were inversely associated with the risk of rheumatoid arthritis and positively associated with the risk of non-rheumatoid arthritis.

Possible mechanisms underlying increased exposure to mosquito repellent and risk of arthritis, studies have shown that exposure to mosquito repellent can affect the expression of peripheral cytokines, leading to increased inflammation [29]. This suggests that DEET may also contribute to the occurrence of arthritis through inflammatory reactions. In addition, mitochondrial dysfunction is closely related to the occurrence of arthritis [30–32]. Recent research reports have shown that DEET exposure can lead to mitochondrial dysfunction in the hippocampus, which is also one of the expected possible mechanisms by which DEET exposure increases the risk of arthritis [33, 34].

Our study found a negative correlation between urinary DCBA levels and the risk of rheumatoid arthritis. Although RA commonly led to disability, work limitations, and higher mortality rates until the 1990s, advancements in therapeutic strategies have since transformed RA into a condition that can be effectively managed [35, 36]. The timely identification of rheumatoid arthritis is a critical aspect in its management due to the distinct treatment approaches required for rheumatoid arthritis compared to other forms of arthritis [37, 38]. Consequently, it is plausible to posit that urinary DCBA levels in U.S. adults may have utility as a biomarker to differentiate and potentially exclude rheumatoid arthritis.

The present study acknowledges several limitations that warrant consideration. The cross-sectional design hinders our capacity to establish causality between DEET exposure and arthritis prevalence, given its snapshot nature and inability to track temporal relationships. Moreover, the absence of data regarding DEET metabolites restricts a more holistic examination of mixed exposures, which could potentially inform the broader implications of DEET on arthritis. Additionally, the reliance on a single measurement of the biomarker DCBA to indicate DEET exposure introduces the possibility of misrepresenting the true variability of long-term

exposure levels, thereby, possibly skewing the actual correlation between DEET and arthritis risk. Longitudinal measurements of DCBA would yield a more comprehensive representation of DEET exposure. Lastly, while our study advances the understanding of the link between DEET exposure and arthritis, the exact biological mechanisms underlying this association are yet to be elucidated. Future research should endeavor to utilize prospective designs and integrate laboratory findings to shed light on the mechanistic pathways at play.

Our research augments the growing compendium of knowledge surrounding the potential health ramifications of DEET exposure, particularly illuminating the nexus between DCBA biomarker concentrations, arthritis, and rheumatoid arthritis. We have identified a noteworthy correlation between exposure to DCBA and the prevalence of arthritis within the adult demographic. Moreover, our data suggests a consequential positive association between escalated DCBA levels and an increased prevalence of arthritis. It is imperative to underscore the necessity for subsequent investigations to corroborate these findings and delve further into the etiological mechanisms at play.

## 5 Conclusion

Our research delineates those elevated concentrations of DCBA correlate positively with a heightened prevalence of arthritis in the adult demographic. In stark contrast, an inverse relationship was observed between elevated DCBA levels and the occurrence of rheumatoid arthritis. Consequently, we posit that DCBA has the prospective utility as a biomarker in the differential diagnosis of arthritis variants.

## Acknowledgments

Our profound gratitude is extended to the committed team at the Centers for Disease Control and Prevention (CDC), as well as the personnel at the National Center for Health Statistics (NCHS), for their indispensable support. We further wish to acknowledge the contributions of the National Health and Nutrition Examination Survey participants, whose participation has been pivotal to the success of this study.

## Author Contributions

**Conceptualization:** Taihong Lv.

**Data curation:** Taihong Lv, Hanming Yu.

**Formal analysis:** Hanming Yu.

**Investigation:** Hanming Yu, Zishuo Ji, Yuru Chen.

**Methodology:** Zishuo Ji, Yuru Chen.

**Project administration:** Li Ma.

**Visualization:** Qi Zhao.

**Writing – original draft:** Taihong Lv, Qi Zhao.

**Writing – review & editing:** Li Ma.

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
