## [Decision Letter · Decision Letter 0]

16 Aug 2024

PONE-D-24-26509Association of N, N-diethyl-m-toluamide (DEET) with arthritis among adult participantsPLOS ONE

Dear Dr. Ma,

Thank you for submitting your manuscript to PLOS ONE. After careful consideration, we feel that it has merit but does not fully meet PLOS ONE’s publication criteria as it currently stands. Therefore, we invite you to submit a revised version of the manuscript that addresses the points raised during the review process.

ACADEMIC EDITOR: After a careful reading of manuscript and following the suggestions of the reviewer I recommend a Major revision of manuscript considering all questions of the reviewer. This decision is justified by PLOS ONEs publication criteria and not for the novelty of perceived impact of your data.

We look forward to receiving your revised manuscript.

Kind regards,

José Luiz Fernandes Vieira

Academic Editor

PLOS ONE

Journal Requirements:

https://doi.org/10.1016/j.chemosphere.2022.135669

In your revision ensure you cite all your sources (including your own works), and quote or rephrase any duplicated text outside the methods section. Further consideration is dependent on these concerns being addressed.

3. Thank you for stating the following in your Competing Interests section:  "None"

Reviewers' comments:

Reviewer's Responses to Questions

**Comments to the Author**

1. Is the manuscript technically sound, and do the data support the conclusions?

Reviewer #1: Yes

2. Has the statistical analysis been performed appropriately and rigorously? 

Reviewer #1: Yes

3. Have the authors made all data underlying the findings in their manuscript fully available?

Reviewer #1: Yes

4. Is the manuscript presented in an intelligible fashion and written in standard English?

Reviewer #1: No

5. Review Comments to the Author

Reviewer #1: 1.NHANES is a cross-sectional survey, and the incidence rate cannot be calculated. In this paper, incidence rate is used to replace the prevalence rate for many times, please check.

2. The result expression is inconsistent with the table content. See Table 2. "Univariate analyses suggest that family income and BMI exhibit no association with sleep disorder prevalence"

3. What is the result of "sleep disorder" on single-factor analysis, as shown in Table 2 on the prevalence of "no association" and on the prevalence of risk factors?

4. The wording of covariates in the table should be consistent in the text

5. Figure 2 and Figure 3 are marked backwards

6. In Figure 2, "non-Mexican Hispanic males between the ages of 45-59", P=0.007, indicating a statistical difference, but OR 95%CI(0.917,5.599) showed no statistical difference, please check

7. Why is there no group of 20-44 years old in this group, please explain or add

8. Why not do other types of grouping? Please explain or add

9. In logistics, Q1 is the reference group, and the reference groups from Q2 to Q4 are all Q1. Q1 does not have P, or P is P for trend

10. The description in table34 does not match the table. Please check and correct it

11. The table is unified, and the data in Q1 is either blank or REF is added

12. The description of quartile is wrong, there is no "quantiles (Q1-Q2)" type, there is only "Q1" quantiles=Q, and the data has been changed into Q1, Q2, Q3 and Q4 in the logistics analysis, which is either a continuous variable or a grouping variable, and there is no expression of Q1-Q2. Please carefully study the significance of data analysis methods and the description of results, and then revise the description of results analysis in the paper.

13.RCS diagram is not clear, please adjust it

14. N, n-diethyl-meta-toluamide is covered at length in the first three paragraphs of the discussion, but is not relevant to this article. The first paragraph is suggested to reduce the length and take the essence of it. It is suggested to shorten these three paragraphs into one paragraph.

15. It is suggested that paragraphs 4, 5 and 6 summarize the results of the whole paper as the beginning of the discussion, emphasizing the innovation and value of this study, but reducing the length and condensing the sentences

16. Increase the comparison between this study and others' studies, as well as possible mechanism explanations for the results found in this study

6. PLOS authors have the option to publish the peer review history of their article (what does this mean?). If published, this will include your full peer review and any attached files.

Reviewer #1: **Yes: **shan liu

---

## [Author Response · Author response to Decision Letter 0]

23 Aug 2024

The content of our response to reviewers is in annex "Response to Reviewers".

---

## [Decision Letter · Decision Letter 1]

17 Oct 2024

Association of N, N-diethyl-m-toluamide (DEET) with arthritis among adult participants

PONE-D-24-26509R1

Dear Dr. Li MA

We’re pleased to inform you that your manuscript has been judged scientifically suitable for publication and will be formally accepted for publication once it meets all outstanding technical requirements.

Kind regards,

José Luiz Fernandes Vieira

Academic Editor

PLOS ONE

Additional Editor Comments (optional):

Reviewers' comments:

Reviewer's Responses to Questions

**Comments to the Author**

1. If the authors have adequately addressed your comments raised in a previous round of review and you feel that this manuscript is now acceptable for publication, you may indicate that here to bypass the “Comments to the Author” section, enter your conflict of interest statement in the “Confidential to Editor” section, and submit your "Accept" recommendation.

Reviewer #1: All comments have been addressed

2. Is the manuscript technically sound, and do the data support the conclusions?

Reviewer #1: Yes

3. Has the statistical analysis been performed appropriately and rigorously? 

Reviewer #1: Yes

4. Have the authors made all data underlying the findings in their manuscript fully available?

Reviewer #1: Yes

5. Is the manuscript presented in an intelligible fashion and written in standard English?

Reviewer #1: Yes

6. Review Comments to the Author

Reviewer #1: (No Response)

7. PLOS authors have the option to publish the peer review history of their article (what does this mean?). If published, this will include your full peer review and any attached files.

Reviewer #1: No

---

## [Editor Report · Acceptance letter]

20 Oct 2024

PONE-D-24-26509R1 

PLOS ONE

Dear Dr. Ma, 

I'm pleased to inform you that your manuscript has been deemed suitable for publication in PLOS ONE. Congratulations! Your manuscript is now being handed over to our production team.

Kind regards, 

on behalf of

Dr. José Luiz Fernandes Vieira 

Academic Editor

PLOS ONE